# Human Intramuscular Hyperimmune Gamma Globulin (hIHGG) Anti-SARS-CoV-2—Characteristics of Intermediates and Final Product

**DOI:** 10.3390/v14061328

**Published:** 2022-06-17

**Authors:** Elzbieta Lachert, Joanna Lasocka, Artur Bielawski, Ewa Sulkowska, Katarzyna Guz, Krzysztof Pyrc, Agnieszka Dabrowska, Agata Wawryniuk-Malmon, Magdalena Letowska, Krzysztof Tomasiewicz, Piotr Grabarczyk

**Affiliations:** 1Department of Transfusion Medicine, Institute of Hematology and Transfusion Medicine, Indiry Gandhi 14 Str., 02-776 Warsaw, Poland; elachert@ihit.waw.pl (E.L.); jlasocka@ihit.waw.pl (J.L.); letowska@ihit.waw.pl (M.L.); 2Biomed Company, Uniwersytecka 10 Str., 20-029 Lublin, Poland; biomed@biomedlublin.com (A.B.); awawryniuk@biomedlublin.com (A.W.-M.); 3Department of Virology, Institute of Hematology and Transfusion Medicine, Chocimska 5 Str., 00-957 Warsaw, Poland; esulkowska@ihit.waw.pl; 4Department of Immunohematology and Transfusion Medicine, Institute of Hematology and Transfusion Medicine, Chocimska 5 Str., 00-957 Warsaw, Poland; kguz@ihit.waw.pl; 5Virogenetics Laboratory of Virology, Malopolska Centre of Biotechnology, Jagiellonian University, Gronostajowa 7A Str., 30-387 Krakow, Poland; k.a.pyrc@uj.edu.pl (K.P.); agnieszka.dabrowska@doctoral.uj.edu.pl (A.D.); 6Microbiology Department, Faculty of Biochemistry, Biophysics and Biotechnology, Jagiellonian University, Gronostajowa 7A Str., 30-387 Krakow, Poland; 7Department of Infectious Diseases, Medical University of Lublin, Stanislawa Staszica 16 Str., 20-081 Lublin, Poland; krzysztoftomasiewicz@umlub.pl

**Keywords:** gamma globulin, SARS-CoV-2, COVID-19, convalescent plasma, human intramuscular hyperimmune gamma globulin anti-SARS-CoV-2 (hIHGG anti-SARS-CoV-2)

## Abstract

This study aims to characterize the intermediates, and the final product (FP) obtained during the production of human intramuscular hyperimmune gamma globulin anti-SARS-CoV-2 (hIHGG anti-SARS-CoV-2) and to determine its stability. **Material and methods**: hIHGG anti-SARS-CoV-2 was fractionated from 270 convalescent plasma donations with the Cohn method. Prior to fractionation, the plasma was inactivated (Theraflex MB Plasma). Samples were defined using enzyme immunoassays (EIA) for anti-S1, anti-RBD S1, and anti-N antibodies, and neutralization assays with SARS-CoV-2 (VN) and pseudoviruses (PVN, decorated with SARS-CoV-2 S protein). Results were expressed as a titer (EIA) or 50% of the neutralization titer (IC_50_) estimated in a four-parameter nonlinear regression model. **Results:** Concentration of anti-S1 antibodies in plasma was similar before and after inactivation. Following fractionation, the anti-S1, anti-RBD, and anti-N (total tests) titers in FP were concentrated approximately 15-fold from 1:4 to 1:63 (1800 BAU/mL), 7-fold from 1:111 to 1:802 and from 1:13 to 1:88, respectively. During production, the IgA (anti-S1) antibody titer was reduced to an undetectable level and the IgM (anti-RBD) titer from 1:115 to 1:24. The neutralizing antibodies (nAb) titer increased in both VN (from 1:40 to 1:160) and PVN (IC_50_ from 63 to 313). The concentration of specific IgG in the FP did not change significantly for 14 months. **Conclusions:** The hIHGG anti-SARS-CoV-2 was stable, with concentration up to approximately 15-fold nAb compared to the source plasma pool.

## 1. Introduction

Much effort has been made to develop effective therapeutic and prophylactic management of COVID-19. The existing therapeutic options were reviewed to combat SARS-CoV-2 infection before effective vaccines and drugs became available. One therapeutic option was a transfusion of convalescent plasma (CP) from SARS-CoV-2 recovered patients (passive immunization). The most effective antibodies in CP exhibiting the highest neutralizing potential bind to the highly immunogenic S protein of the virus (S1-RBD, receptor binding domain in S1 subunit within spike protein) and inhibit the interaction of the spike protein with the host cell receptor, the angiotensin-converting enzyme 2 (ACE2) or the entry process. Other specificity antibodies (e.g., directed against nucleocapsid protein) may also have a neutralizing potential, but this effect is less pronounced [1]. Their protective effect is restricted to opsonization and macrophage activation of the cellular response and phagocytosis [2]. Until 2021, high-titer antibody-neutralizing CPs were transfused in the early stage of SARS-CoV-2 infection, with unconfirmed therapeutic outcomes [3,4]. No apparent clinical efficacy may be attributed to the lack of standardization of CP units, which differ with regard to antibody specificity, isotypes, and titer [5]. Some recent observations, however, show that high titer convalescent plasma may be an effective early outpatient COVID-19 treatment with the advantages of low cost, wide availability, and rapid resilience to variant emergence from viral genetic drift in the face of a changing pandemic [6]. Another promising therapeutic option was the administration of specific monoclonal antibodies in the early infection phase [7]. Finally, intravenous or intramuscular human hyperimmune gamma globulin anti-SARS-CoV-2 is considered the reasonable alternative to unstandardized CP and specific monoclonal antibodies, the production of which is complicated and costly. As yet, there is no sufficient experience with this kind of immunological therapy or prophylaxis, and well-documented publications on the subject are scarce [8,9]. In Poland, a specific intramuscular human hyperimmune gamma globulin anti-SARS-CoV-2 (hIHGG anti-SARS-CoV-2) was produced and assessed in a clinical trial conducted from December 2020 to December 2021 within a project financed by the Medical Research Agency.

The aim was to characterize plasma for fractionation and the intermediates obtained during the production of hIHGG anti-SARS-CoV-2. Characteristics of the final product (FP) and the results of product stability studies are also presented.

## 2. Material and Methods

The batch of hIHGG anti-SARS-CoV-2 was produced from 270 units of fresh frozen plasma (FFP) collected from 194 Polish donors during the first months of the pandemic (1 June 2020–17 August 2020). The source plasma from automated plasmapheresis was supplied by Blood Transfusion Centers. Donors qualified according to Polish legal regulations were recruited from among COVID-19 convalescents after a minimum of 14 days of the repeat negative NAT test or after at least 28 days of symptom disappearance or termination of (post-discharge) quarantine [10,11,12].

Appendix A in the Appendix A presents the distribution of ABO blood type among donors whose plasma was used for hIHGG anti-SARS-CoV-2 production.

Prior to pooling, all FFP (CP) units were thawed (35 ± 2 °C, 20 min) and subjected to pathogen inactivation with methylene blue (Theraflex MB Plasma system, Macopharma, Tourcoing, France).

### 2.1. Production of hIHGG Anti-SARS-CoV-2

hIHGG anti-SARS-CoV-2 was manufactured for the first time by Biomed Lublin S.A. using the technique based on plasma fractionation with cold ethanol (Cohn’s procedure) [13]. The method consists of multiple selective precipitations of plasma proteins, using different ethanol concentrations, pH, temperature, and ionic strength. Plasma proteins are heat-inactivated, so the process proceeds within the 0–10 °C range. The active substance (lyophilized anti-SARS-CoV-2 immunoglobulin with a determined level of anti-SARS-CoV-2 antibodies) was suspended in a solution of glycine and 0.3% sodium chloride. The resulting product was filtered for clarification and sterilization. The FP was automatically packaged into 4000 (2 mL) ampoules. 

IgG, IgA, IgM, and neutralizing antibody titers were measured in plasma before and after inactivation, both in the FP and the intermediates: 270-unit plasma pool (A), concentrate after fractionation but before lyophilization (B), the FP before filtration and ampoule-filling (C) and the FP—10% hIHGG anti-SARS-CoV-2, lot 0120 (D) (Figure 1).

### 2.2. Methods of Measuring Anti-SARS-CoV-2 Antibodies

#### 2.2.1. Immunoenzymatic Methods

The following tests were used for the detection of specific anti-SARS-CoV-2 antibodies in the intermediates and the FP:SARS-CoV-2 Ab ELISA Test (Wantai, Beijing, China)—for simultaneous detection of three antibody isotypes (IgM, IgA, IgG) directed to the S1-RBD;SARS-CoV-2 Ab IgM ELISA (Wantai, Beijing, China)—for detecting the IgM isotype of specific antibodies directed to the S1-RBD;Anti-SARS-CoV-2 IgA ELISA (EUROIMMUN, Lübeck, Germany)—for semi-quantitative in vitro evaluation of anti-SARS-CoV-2 IgA antibodies directed to the S1 protein;Anti-SARS-CoV-2 ELISA (IgG) (EUROIMMUN, Lübeck, Germany)—for semi-quantitative in vitro evaluation of anti-SARS-CoV-2 IgG antibodies directed to the S1 protein. The test was used in 2020, and since December 2020, anti-SARS-CoV-2 QuantiVac ELISA (IgG) (EUROIMMUN, Lübeck, Germany) quantitative test was in use;iFlash–SARS-CoV-2 IgG—chemiluminescence test for semi-quantitative detection of anti-SARS-CoV-2 IgG antibodies (SHENZHEN YHLO BIOTECH CO., Shenzhen, China);Bio-Rad Platelia SARS-CoV-2 Total Ab—for simultaneous detection of IgG, IgA, and IgM antibodies directed to the nucleocapsid (anti-N).

#### 2.2.2. Neutralization Tests

Characteristics of anti-SARS-CoV-2 antibodies were based on SARS-CoV-2 virus neutralization assay (VN) and pseudovirus assay (PVN) to determine SARS-CoV-2 neutralizing activity. Details concerning assays and testing procedures are presented in the Appendix A.

#### 2.2.3. Quantitative and Semi-Quantitative Antibody Concentration Assay

A semi-quantitative antibody concentration assay in the intermediates and the FP was performed as described above. The dilutions with Phosphate Buffered Saline solution (PBS) were from 1:20 to 1:5120. A titer-value twice lower or higher than the reference was considered statistically significant.

#### 2.2.4. Stability Studies

Between September 2020 and December 2021, the stability of hIHGG anti-SARS-CoV-2 stored at 2–8 °C was assessed monthly, based on the measurements of anti-S1 RBD ELISA Test (Wantai, Beijing, China) and IgG antibodies directed against S1 protein (EUROIMMUN, Lübeck, Germany). At some time points, SARS-CoV-2 virus neutralization (VN) was performed.

#### 2.2.5. Statistical Analysis

Statistica 13.3 (Tibco, Statsoft, Poland v. 13.3) or RStudio software (RStudio ver, 1.4.1106, Boston USA)were used for statistical analyses and figure presentation. Four-parameter non-linear regression method was used for IC_50_ estimation and assessment of anti-SARS-CoV-2 titer values in immunoassays.

## 3. Results

### 3.1. Characteristics of Plasma Used for the Production of hIHGG Anti-SARS-CoV-2

Anti-S1-RBD tests (total test) were positive for 268 out of 270 (99.26%) of CP units and negative for 2 (0.74%). In 158 samples (59%), the S/Co value was the highest (>19.9) (Figure 2a). The iFlash S1 SARS-CoV-2 IgG antibody test was positive in 183 out of 206 (88.83%) CP units and negative in 22 (11.16%). In 88 plasma units (42.71%) the antibody titer was >39.99 AU (arbitrary units)/mL. The low reactivity (9.99–39.99 AU/mL) was detected in 46.12% of units. Only four (1.9%) plasma units presented a very high IgG antibody titer (>129.99 AU/mL) (Figure 2b). A similar distribution was obtained with the Euroimmun assay designed for the semi-quantitative analysis of IgG antibodies. A positive result in the anti-S1 IgG test was obtained in 92.07% CP tested units and a negative result was found for 3.77%, while 4.15% of plasma units were in the gray zone. An S/Co value above 2.99 was determined in 43.40% of source plasma units, while the lowest reactivity (1.10–2.99 S/Co) was measured in 49% of units. As with the iFlash test, a small percentage (3.77%) of the samples presented the highest reactivity (S/Co > 7.99) (Figure 2c). The frequency of positive results in iFlash and Euroimmun tests did not differ (*p* = 0.23) but was higher (*p* < 0.05) in the Wantai test. A positive result in the anti-S1 IgA test was obtained in 75.14% of CP units and a negative result was obtained in 10.17%; 14.69% units were in the gray zone (0.80–1.10 S/Co). In 47 plasma units (26.55%) the S/Co value was >2.55, and reactivity was the lowest (1.10–2.55 S/Co) in 86.49% of units. Only 5.08% of units presented the highest reactivity (>7.5 S/Co) (Figure 2d). Additionally, the iFlash test was used to determine anti-SARS-CoV-2 IgG antibodies in plasma after pathogen inactivation. The titer of specific antibodies before and after inactivation did not differ (Figure 3).

### 3.2. Characteristics of Intermediates

Results of enzyme immunoassays and neutralization tests in dilutions of intermediates are presented in Appendix A. Table 1 shows the characteristics of the intermediates (A-D) in terms of antibody titers and PVN IC_50_. The parameters were estimated using a four-parameter regression curve. Experimentally determined titers of neutralizing antibodies (nAb) were analyzed for VN.

A.The plasma pool

The anti-S1-RBD and anti-N total antibody tests’ reactivity was observed in the plasma pool allowed to detect of IgG, IgM, and IgA to the S1-RBD and the nucleocapsid, respectively. IgG and IgA anti-S1 antibodies and IgM anti-RBD S1 antibodies were also detected. Table 1 presents antibody titers of different specificity and neutralizing antibodies (nAb). Complete inhibition of the cytopathic effect in VN was observed for the titer 1:40, and partial inhibition also at higher dilutions, up to 1:1280. In the plasma pool, 60.62 AU/mL was found (iFlash method); 71.61 RU/mL (Euroimmun assay) corresponded to 229.15 BAU/mL (anti-S1 IgG antibody assay, Appendix A).

B.Concentrate after fractionation, before lyophilization

Following fractionation, an increase in anti-S1 RBD and the anti-N antibody titer was reported in total tests (approximately 4- and >6-fold, respectively) as a result of an approximately 8-fold higher concentration of IgG and an almost 6-fold reduction in IgM titer. The iFlash anti-S1 IgG antibody titer in the concentrate was 163.14 AU/mL and 1109.12 BAU/mL in ELISA (Euroimmun). Compared to the plasma pool, the neutralizing antibody titer (nAb) after fractionation increased four-fold to 160 (complete inhibition of the cytopathic effect). Partial inhibition was also observed at higher dilutions (up to 1:1280) of the concentrate (Appendix A).

From the concentrate obtained from fractionation to select the best option for the FP, 10% and 15% concentrations of immunoglobulin solutions were prepared. No significant differences were observed between the 10% and 15% concentrations, so a 10% concentration was chosen. No anti-S1 IgA antibodies were detected in the intermediates except the plasma pool (Table 1), while the high neutralizing activity of both solutions (10% and 15%) was observed. Complete inhibition of the cytopathic effect was determined up to a dilution of 1:1280, and partial inhibition even up to 1:5120 (Appendix A).

### 3.3. Testing the Antibodies in the FP before © and after Ampoule Filling (D)

FP reactivity before and after ampoule filling in specific EIA tests is presented in Table 1. Antibody titer was concentrated in FP as compared to the source plasma pool; the anti-S1 titer increased 15-fold (up to 1566–1765 BAU/mL) from 1:4.2 in the production pool (PP) to 1:63 in the final FP, and anti-RBD and anti-N concentrations (total tests) increased approximately seven-fold; from 1:111 to 1:802 and from 1:13 to 1:88, respectively. During production, the IgA (anti-S1) antibody titer was reduced to an undetectable level and IgM (anti-RBD) titer from 1:115 in PP to 1:24 in FP. A four- to five-fold increase in neutralizing antibody titer was observed in VN (from 1:40 in PP to 1:160 in FP) and PVN (IC_50_ from 63 in PP to 313 in FP).

Antibody specificity, isotype, and titer in FP did not differ from the solution prior to ampoule filling (10%). FP, however, demonstrated a complete inhibition of the cytopathic effect up to a dilution of 1:160 and partial inhibition effect to a dilution of 1:2560 (Appendix A).

### 3.4. Physico-Chemical Characteristics of hIHGG Anti-SARS-CoV-2

hIHGG anti-SARS-CoV-2 solution is a clear, colorless liquid with a pH of 6.3. Anti-HBs antibodies (81 IU/g of immunoglobulin) were found in the sterile product. In molecular weight distribution, the product’s main component corresponded to the IgG in normal human serum. Relative retention for monomer and dimer peak in the chromatogram of the reference solution was 1; the sum of the monomer and dimer peak areas of the total chromatogram covered 95%, and the sum of the polymer peak areas and aggregates of the total chromatogram area was 4%. No more than 10% of the protein differs in mobility from the main bands.

### 3.5. Product Stability

hIHGG anti-SARS-CoV-2 stability results (lot 0120) are presented in Table 2. They varied during the study period, but no clear trend was observed. Moreover, the reduction in the specific anti-SARS-CoV-2 antibodies and nAb titer never exceeded two, as compared to the FP obtained in September 2020 (Table 2). This confirms the stability of the antibody titer in hIHGG anti-SARS-CoV-2 over 14 months. The differences between results may be due to variability of test reactivity.

Inhibition of the cytopathic effect with hIHGG anti-SARS-CoV-2 was determined using the VN test at three time points: in October and November 2020 and in March 2021. The complete (100%) protective effect was observed after adding 160-fold dilution of hIHGG anti-SARS-CoV-2 to the infected cell culture. Partial inhibition of cytopathic development was observed up to a dilution of 1:2560 (October and November 2020) or up to 1:1280 (March 2021).

## 4. Discussion

The Biomed Lublin Serum and Vaccine Factory used the same technology to produce hIHGG anti-SARS-CoV-2 and to manufacture specific immunoglobulins for prophylactic and therapeutic purposes (e.g., GAMMA anti-HBs, GAMMA anti-D). For years now, GAMMA anti-HBs have been used for HBV prophylaxis after potential exposure to the virus. Previously, immunological treatment with CP was reported during the Spanish flu pandemic in 1918 for CMV or parvovirus B19 infections, measles, mumps, varicella, coronavirus-induced Severe Acute Respiratory Syndrome (SARS), and Middle East Respiratory Syndrome (MERS) [14,15,16]. Administered to immunocompromised patients or at the onset of infection (the humoral immune response has not yet occurred), these products hasten recovery or moderate the course of illness [17]. Unlike plasma transfusions, hIHGG anti-SARS-CoV-2 is not expected to induce severe adverse reactions, allergic reactions, transfusion-related acute lung injury (TRALI), and circulatory overload (TACO) [18]. Moreover, CP transfusions with large amounts of complementary proteins and coagulation factors put these patients at high risk of thromboembolic complications. With purified immunoglobulin preparations, the risk is lower because complementary proteins and coagulation factors are absent compared to CP [19,20].

According to O’Brien et al. [21], several countries are planning or have started the production of anti-SARS-CoV-2 hyperimmune immunoglobulin. Literature reports on the production of immunoglobulin from CP are still scarce. One comes from Pakistan, where human intravenous hyperimmune anti-SARS-CoV-2 immunoglobulin was prepared from plasma pools of 16–20 donations using precipitation with caprylic acid. The concentration of IgG anti-SARS-CoV-2 antibodies in the FP was approximately three-fold higher than in the plasma pool [22]. Vandeberg et al. used the chromatographic method to obtain hyperimmunized immunoglobulin and reached a 10-fold higher concentration of anti-SARS-CoV-2 antibodies in the FP than in the CP pool [23]. In our study, the concentration of specific IgG antibodies in FP was eight-fold higher, IgA antibodies were reduced to an undetectable level, and IgM antibody concentration was almost six times lower. As with Vanderberg’s study, in our FP, the antibody neutralizing activity in the VN assay was approximately four- to five-fold higher than in the plasma pool (160). During therapy or prophylaxis with intravenous or intramuscular human hyperimmune gamma globulin anti-SARS-CoV-2, the patient receives antibodies of higher-neutralizing activity in a much-reduced volume compared to CP. CP fractionation effectively reduces IgM and IgA antibody isotypes, and practically only IgG remains in the FP. It is essential to eliminate IgM-class antibodies as they are responsible for anti-A and anti-B antibody-mediated intravascular hemolysis [24]; however, the loss of the IgA isotype may weaken the protective potential of the plasma-derived product tested in current study [25]. Hyperimmunized immunoglobulin has the additional advantage of inhibiting transmission of potentially harmful clotting factors transfused with FFP from convalescent donors. It also allows accurate dosage of antibody concentration. It is worth underlining that the studies above focused mainly on the characteristics of the FP. In our study, we not only characterized the FP (Gamma anti-SARS-CoV-2 (10%), series 0120) but also assessed the quality of individual plasma units, the plasma pool, the concentrate obtained after fractionation and before lyophilization, and also the FP before ampoule filling (prior to filtration).

It is noteworthy that COVID-19 convalescents mainly were first-time donors, generally believed to be less safe than repeat donors due to the higher prevalence of blood-borne pathogens [26,27]. Their plasma was not subjected to quarantine but was pathogen-inactivated (Theraflex MB Plasma system) after thawing but before fractionation to strengthen the safety profile. Methylene blue (MB) enhanced singlet oxygen production, altering the protein structure. In our study, we first examined the effect of inactivation on the antibody titer. Raster et al. and Kostin et al. demonstrated that inactivation with MB did not inhibit IgM and IgG affinity of binding their epitopes or the interaction of IgG with the Fc receptor [28,29]. This explains the lack of difference in the iFlash test for the concentration of specific plasma antibodies before and after inactivation.

When in June–August 2020 Biomed, Lublin was supplied with CP for production of hIHGG anti-SARS-CoV-2, they were in no position to decide about the antibody titer. Implementation of mass vaccination and successive waves of infections changed the situation radically. By the end of 2021, almost 85% of donors at the Warsaw BTC were anti-RBD S1 positive, and >65% were donors with high-specific-Ig load (S/Co > 19.9; IHTM data). We may, therefore, expect hIHGG anti-SARS-CoV-2 manufactured from recently collected plasma to have higher titer antibodies and prove more effective in a clinical setting. Plasma collected two years after the pandemic outbreak may have higher neutralizing potential against different variants of the virus than the plasma from the first months of the epidemic.

Our study demonstrates that both the hIHGG production technology and product protocol (the medicinal product, CP, and intermediates) can relatively quickly be adapted to the reality of new pandemics. The described procedure may be considered an essential trial before the emergence of any further SARS-CoV-2 variants or any other new infectious agents.

The studies on the therapeutic effectiveness and safety of COVID-19 therapy with immunoglobulin have been completed, and the results will be published soon. Another point worth considering is the application of the product for prophylactic use (passive immunotherapy), especially for immunosuppressed patients.

## Figures and Tables

**Figure 1 viruses-14-01328-f001:**
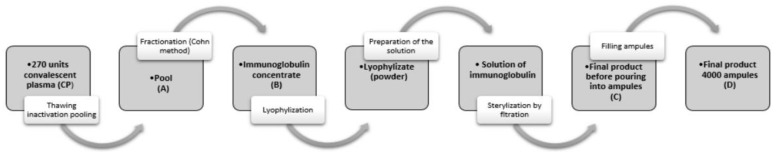
hIHGG anti-SARS-CoV-2 production.

**Figure 2 viruses-14-01328-f002:**
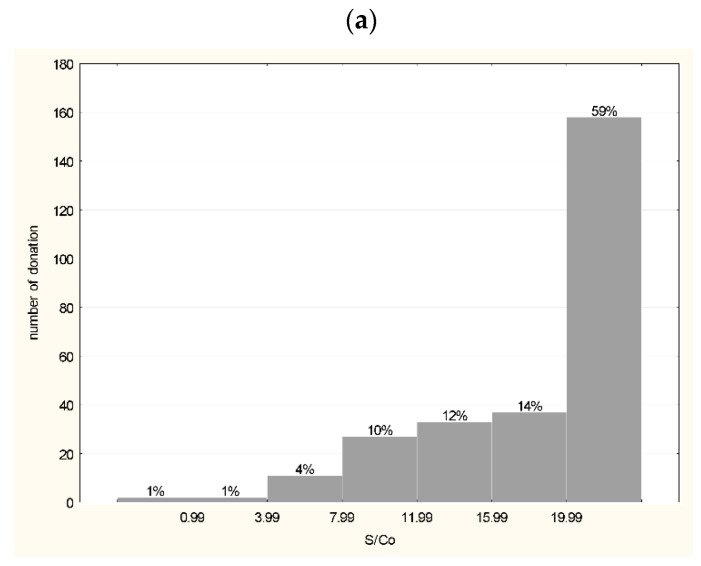
Histogram of anti-SARS-CoV-2 antibodies reactivity in EIA assays in convalescent plasma units used for the production of hIHGG anti-SARS-CoV-2: (**a**) Wantai anti-S1 RBD Ab Total—IgM, IgA and IgG (n = 270, S/Co < 1 negative result, S/Co > 19.9 over result); (**b**) iFlash anti-S1 IgG (n = 206, <10 AU/mL—negative); (**c**) Euroimmun anti-S1 IgG (n = 265, S/Co < 0.8 negative, 0.8–1.1 gray zone, >1.1 positive); (**d**) Euroimmun IgA (n = 177, S/Co < 0.8 negative, 0.8–1.1 gray zone, >1.1 positive). Above bars % of tested samples is presented.

**Figure 3 viruses-14-01328-f003:**
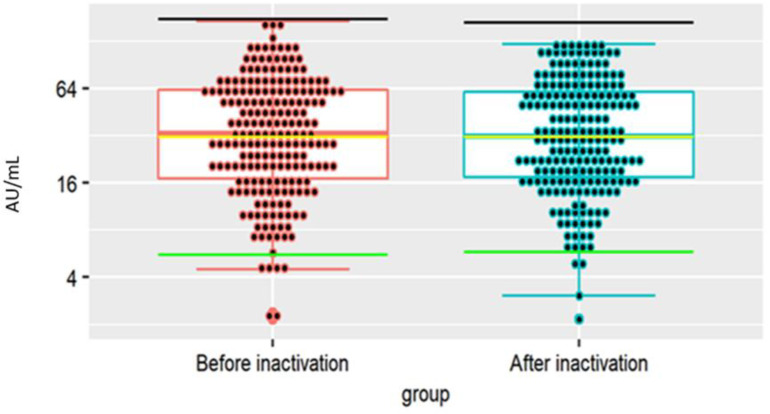
Comparison of iFlash test reactivity (anti-S1 SARS-CoV-2 IgG antibody assay) in 206 CP donations before and after inactivation—no statistically significant differences in the pairwise Wilcoxon test were observed, *p* = 0.502.

**Table 1 viruses-14-01328-t001:** Anti-SARS-CoV-2 antibody titer of different specificity and isotypes in the intermediates and in the FP.

Test/Assay	Production Stage				Compaction Factor
	Plasma Pool	Concentrate Prior to Liofilization	Product		
			Before Filling	After Filling	
	A	B	C	D	D/A
anti-RBD S1 total (Wantai)	110.90	471.04	721.28	801.84	7.23
anti-S1 IgG (EuroImmun)	4.21	33.34	61.87	63.31	15.02
anti-S1 IgA (EuroImmun)	1	undetectable	undetectable	undetectable	-
anti-RBD S1 IgM (Wantai)	115.52	20	21.14	24.43	0.21
anti-N total (BioRad)	13.99	83.61	123.97	87.57	6.74
Neutralization VN	40	160	nt.	160	4
PVN (IC_50_)	63.71	nt.	nt.	313.55	4.92

nt.—not tested. 4.72-fold dilution.

**Table 2 viruses-14-01328-t002:** Results of stability tests of GAMMA anti-SARS-CoV-2 globulin (hIHGG anti-SARS-CoV-2, lot 0120) detecting specific anti-S1 RBD antibodies (total IgG, IgA and IgM), anti-S1 IgG and neutralizing antibodies (neutralization test). Results presented as titers (except VN) were estimated by the four-parameter non-linear regression method. nt: not tested.

	Month													
	1	2	3	4	5	6	7	8	9	10	11	12	13	14
Anti-RBD S1 total Wantai (titer)		981	899	503	nt	801	nt	868	640	nt	680	827	777	960
Anti-S1(IgG) Euroimmun (semiquantitative; titer)	55	49	47											
Anti-S1(IgG) Euroimmun (quantitative) BAU/mL		1765		1653	1667	1566	1606	2104	1855	1798	1833	1953	1650	1765
Neutralization assay (VN) titer at 100% neutralization	160	160				160								

## Data Availability

The Medical Research Agency (MRA).

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
