# Peer review of "Human Intramuscular Hyperimmune Gamma Globulin (hIHGG) Anti-SARS-CoV-2—Characteristics of Intermediates and Final Product"

_viruses, 2022, doi:10.3390/v14061328_

Round 1

Reviewer 1 Report

The authors report the results of their study to determine the neutralization activity of hyperimmune immunoglobulin against SARS-CoV-2. The premise of this study is of some significance, since these blood products may be used for potential treatment of COVID-19 in certain settings. This is a straight-forward and thorough study of modest impact, and the authors should be commended for their rigorous assessment.

The authors have clearly established that their preparation method for hyperimmune serum enriches for immunoglobulins with effective neutralization activity against both live virus and pseudovirus assays, albeit while depleting for certain potentially important antibody species found in unenriched convalescent serum. Notably, they report the expected depletion of IgA from the product- although this clearly does not fully impair in vitro neutralization activity mediated by IgG, it may have some impact on the therapeutic potential of hyperimmune IgG in vivo. I would like the authors to acknowledge this potential limitation.

Author Response

Thank you very much for your review-contribution. All your comments have been acknowledged and introduced into the text. Please see the attachment.

Reviewer 2 Report

This study aims to characterize the intermediates, and the final product (FP) obtained during the production of human intramuscular hyperimmune gamma globulin anti-SARS-CoV-2 (hIHGG anti-SARS-COV-2) and to determine its stability. The work is interesting however some grammatical and scientific  revisions should be necessary. Like for example in an assay. Ex: The sample was diluted 1/20 to 1:2480 etc... .It not necessary to described all the title! An important points as the concentration of anti-S1 antibodies in plasma was similar before and after inactivation. Was the antibody purification performed by the group for the first time ? The concentration of specific antibodies in the FP did not change significantly during 14 months. . It is related to IgG?

In Poland, a specific intramuscular human hyperimmune gamma globulin anti-SARS-CoV-2 (hIHGG anti-SARS-CoV-2) was produced andassessed in a clinical trial within a project financed by the Medical Research Agency.  When started therapy? 

The method consists of multiple selective precipitations of plasma proteins, using different ethanol concentrations, pH,temperature, and ionic strength. Plasma proteins are heat-inactivated, so the process proceeds within the 0-10°C range. Reference of the technology used and more details is required mainly related to immunoglobulin classes!

Author Response

Thank you much for your review-contribution. All your comments have been acknowledged and introduced into the text. Please see the attachment.
